# Nitrogen dioxide pollution in 346 Chinese cities: Spatiotemporal variations and natural drivers from multi-source remote sensing data

Feifei Cheng[1☯], Qing Sun[2☯], Jiaqi Zhang[1], Jiahe Liu[2], Wenwen Lü[1], Fapeng Shen[2], Heming Yang[1*]

1 Jilin Normal University, Siping, Jilin, China, 2 Jilin Agricultural University, Changchun, Jilin, China

☯ These authors contributed equally to this study.
* wosyhm@jlnu.edu.cn

## Abstract

In this study, tropospheric column concentration of nitrogen dioxide ($TNO_2CC$) were derived from Sentinel-5P data. We employed statistical and local spatial autocorrelation analyses to investigate the spatialtemporal distribution and variation of $TNO_2CC$ across 346 major Chinese cities from 2019 to 2023. Using Random Forest (RF) and Shapley Additive Explanations (SHAP), we analyzed the influence of 15 natural factors on ambient $TNO_2CC$ levels. The high $R^2$ values (0.92 and 0.76), along with the close adherence to the 1:1 line, demonstrate the model's robustness. The most influential natural factors identified include atmospheric pressure, aerosol optical depth, Leaf Area Index, evapotranspiration, and dew point temperature. Additionally, a non-linear response curve approach was applied to examine the independent association between natural driving factors and pollutant concentrations. $TNO_2CC$ varied seasonally across the 346 cities, with the highest levels in winter and the lowest in summer. From 2019 to 2023, $TNO_2CC$ levels exhibited fluctuating trends, with notable regional disparities: higher concentrations were observed in capital cities and in northern and northeastern part of China. $TNO_2CC$ were significantly influenced by temperature-related variables, aerosol optical depth, and leaf area index. The findings of this study identify key natural influencing factors and provide a scientific basis for revealing the causes of urban air pollution in China, informing pollution control strategies, identifying priority areas for remediation, and supporting the natural formulation of protection policies.

## Introduction

In recent years, environmental problems caused by air pollutants have attracted increasing attention [1]. Air pollution, which threatens human health and ecosystems, is a global environmental issue [2]. Air quality in any region is directly influenced by

**Data availability statement:** All raw data files are available from the figshare database at DOI: 10.6084/m9.figshare.29948741.

**Funding:** This research was supported by the Science and Technology Department of Jilin Province. Approval No.: YDZJ202301ZYTS234 (received by Drs. Heming Yang, Feifei Cheng, Jiaqi Zhang, and Wenwen Lü) and YDZJ202401520ZYTS (received by Drs. Heming Yang, Qing Sun, Fapeng Shen, and Jiahe Liu).

**Competing interests:** The authors have declared that no competing interests exist.

local human activities [3].Nitrogen dioxide ($NO_2$) is both a pollutant in its own right and a precursor to other pollutants, such as $O_3$ through photochemical reactions with volatile organic compounds (VOCs) and fine particulate mater ($PM_{2.5}$) via nitrate formation [4]. $TNO_2CC$ is an important indicator of air pollution [5], as it contributes to acid rain, acid fog, and photochemical smog [6], increases $PM_{2.5}$ concentrations [7–8], threatens public health [9–12], and harms both society and the ecological environment [13]. Therefore, it is essential to study $NO_2$ pollution [14].

To systematically investigate the etiology of atmospheric pollution and develop source-oriented mitigation strategies, scholars have predominantly focused on two key research avenues: (1) the spatiotemporal variations of air pollution, and (2) the identification of multi-scale drivers influencing pollutant dynamics [15]. Traditional ground-based air pollution monitoring and analysis method are often limited due to (1) insufficient sampling across spatial and temporal dimensions [16], and (2) the uncertainties associated with interference from other gases [17]. Random Forest (RF), a common Machine Learning (ML) method, effectively identifies key features from high-dimensional datasets, making it a robust tool for pollution analysis [18].

Previous studies have identified various factors that may influence air pollution. For example, Wang et al. (2015) observed strong correlations between atmospheric pollutants and meteorological parameters, such as temperature, relative humidity, wind speed, and precipitation, in Wuxi's urban area during 2014 [19]. Liu et al. reported a significant positive correlation between the level of urbanization, human activity intensity, and environmental pollutant concentrations in Fangshan District, Beijing [20]. Additional factors, such as vegetation coverage index [21], have also been widely examined by numerous researchers [22–30]. These influencing factors are widely recognized. Building on this foundation, 15 independent variables were selected to study $TNO_2CC$ based on the principles of data representativeness, quantifiability, and accessibility, while accounting for the systemic interrelationships among multiple factors. The variables include: Pressure (atmospheric pressure), LAI (Leaf Area Index), Dew (dew point temperature), AOD (aerosol optical depth), WS (wind speed), Month (month), ET (evapotranspiration), RH (relative humidity), Pre (precipitation), Fire (fire activity), LST (land surface temperature), Temp (air temperature), GPP (gross primary productivity), Year (year), and Snow (snow cover).

The Tropospheric Monitoring Instrument (TROPOMI) effectively observes global atmospheric trace gases, including $NO_2$. This study utilizes the Google Earth Engine (GEE) platform to retrieve $TNO_2CC$ data, integrating a particle swarm optimization-enhanced random forest with Shapley Additive Explanations (SHAP) interpretation algorithms to systematically analyze the spatiotemporal variation patterns and natural driving factors of $NO_2$ pollution across China's prefecture-level cities. Within this framework, we quantitatively examine the impacts of meteorological drivers on regional air quality dynamics. The integration of satellite remote sensing and explainable machine learning offers a robust analytical framework for identifying pollution control priorities, thereby establishing a theoretical foundation for evidence-based atmospheric pollution mitigation strategies in China.

## Materials and methods

### Study area

As the world's largest developing country, China covers vast territorial expanses and encompasses diverse natural environments. The nation extends across extensive latitudinal gradients and traverses multiple climatic zones, including tropical, subtropical, temperate, and frigid zones. These distinct climatic regimes exhibit significant variations in temperature, humidity, and wind patterns. For instance, southern cities experience humid subtropical conditions, while northern regions are characterized by continental aridity. China's landforms are complex and diverse, comprising plateaus, mountains, plains, basins, and other landform types. Mountainous and plateau regions facilitate air mass exchange, whereas basin and plain terrains restrict pollutant diffusion. As shown in Fig 1, urban development patterns vary markedly across Chinese 346 study units, which include sub-provincial cities, prefecture-level municipalities, and autonomous prefectures. These urban centers, predominantly concentrated in eastern China, display divergent industrialization trajectories, urbanization rates, and population densities. The selected cities encapsulate representative variations in terrain, climatic, and environmental governance frameworks, ensuring a comprehensive analysis of regional atmospheric dynamics.

Hu Huanyong Line(or Hu Line)is a demographic and geographic dividing line in China, proposed by Chinese geographer Hu Huanyong in 1935. It illustrates the uneven distribution of China's population and economic activity from northeast to southwest.The administrative division data in GeoJSON format is sourced from the National Geospatial Information Public Service Platform (Tianditu), with the website: https://cloudcenter.tianditu.gov.cn/administrativeDivision. The map approval number is GS (2024) 0650. The data coordinates are in GCS_WGS_1984. Global Artificial Impervious

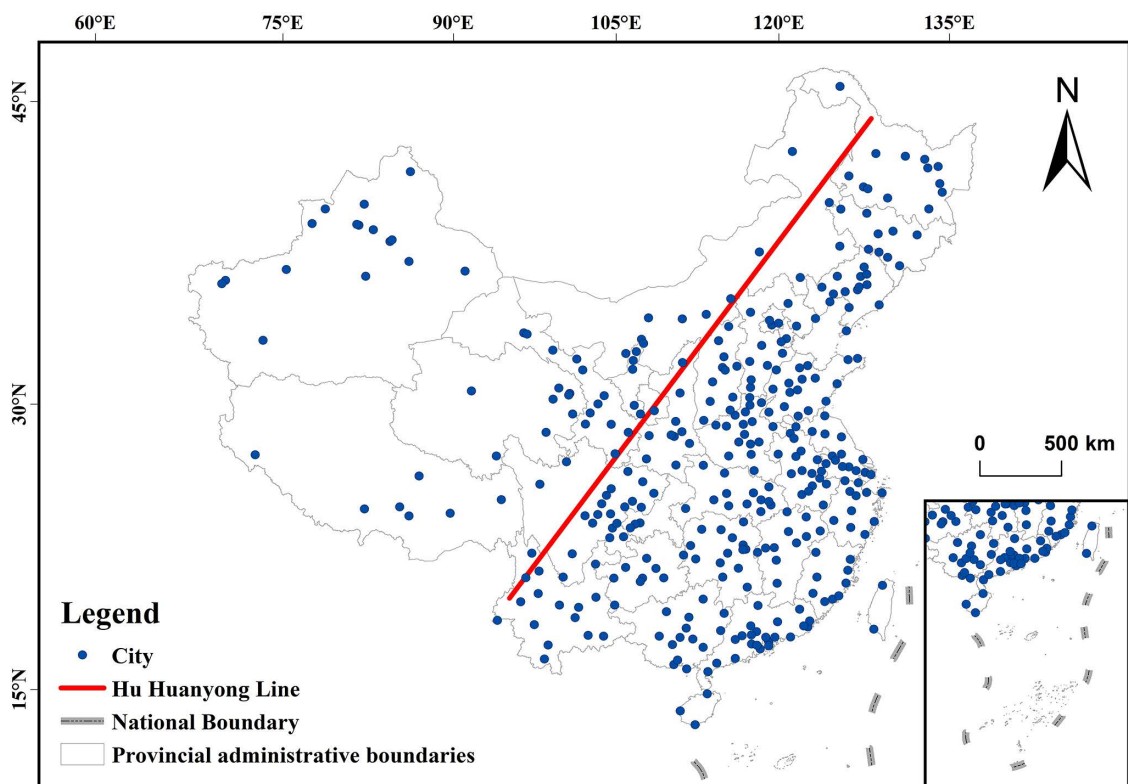

**Fig 1. Distribution map of major cities in China.**

Area (GAIA) dataset, Version 10 (v10). Developed by: Tsinghua University/Fine Resolution Observation and Monitoring of Global Land Cover (FROM-GLC). http://data.ess.tsinghua.edu.cn

## Data sources

The Sentinel satellite series is a component of the European Copernicus Programme. The Sentinel-5P (S5P) satellite, launched in 2017, is designed for real-time monitoring of various atmospheric trace gases, aerosols, and cloud distributions on a global sacale (https://dataspace.copernicus.eu/). Since August 2019, it has achieved a minimum spatial resolution of 5.5×3.5 kilometers. The main data products include Level 1B, which consists of radiometrically calibrated raw spectral data; Level 2: which provides retrieved vertical column concentrations of atmospheric components such as the $TNO_2CC$; and Level 3, which offers gridded data after spatiotemporal aggregation, such as global daily/monthly averages. The Level 3 products deliver gridded daily averages measured by the TROPOMI spectrometer. As measurements are taken at same time each day (early afternoon), these data effectively capture the $NO_2$ pollution characteristics of urban agglomerations in China.

This study uses the OFFL products of Sentinel-5P for research, and uses the "COPERNICUS_S5P_OFFL_L3_NO_2" dataset to count $TNO_2CC$, which includes annual, seasonal, and monthly $TNO_2CC$ data from 2019 to 2023. The OFFL_L3 product is selected because its update frequency is 1–2 months, which is suitable for scientific research and long-term trend analysis, and the data has undergone complete radiometric calibration, spectral correction, and optimization of retrieval algorithms (such as the update of the TM5 meteorological model) to reduce the impact of cloud cover, etc. In contrast, NRT (Near-Real-Time) data is updated hourly and is mainly used for real-time monitoring. More importantly, the Level 3 data has been aggregated into gridded products (such as 0.01° resolution), which directly supports the calculation of seasonal/annual averages, while NRT requires users to process it themselves. All satellite nitrogen dioxide data used in this study are tropospheric column concentration, with a unit of mol/m². Other data sources are shown in Table 1:

ECMWF/ERA5_LAND/MONTHLY: The ERE5-LAND-MONTHLY dataset available on the Google Earth Engine (GEE) platform combines model data with measured data from countries around the world using physical laws, with a spatial resolution of 10,000 meters and a temporal resolution of 1 month. In this study, bands such as 'temperature_2m','total_precipitation', 'dewpoint_temperature_2m','surface_pressure','u_component_of_wind_10 m'and'v_component_of_wind_10 m'from the "ECMWF/ERA5_LAND/MONTHLY" dataset in GEE are used for the statistics of meteorological data. https://doi.org/10.1038/s41558-024-02035-w

**Table 1. Data source for the study area.**

| Date | Source |
|---|---|
| Temp | ECMWF/ERA5_LAND/MONTHLY |
| LST | MODIS/061/MOD11A2 |
| Pressure | ECMWF/ERA5_LAND/MONTHLY |
| WS | ECMWF/ERA5_LAND/MONTHLY |
| RH | ECMWF/ERA5_LAND/MONTHLY |
| Dew | ECMWF/ERA5_LAND/MONTHLY |
| Pre | ECMWF/ERA5_LAND/MONTHLY |
| ET | MODIS/006/MOD16A2 |
| Snow | MODIS/061/MOD10A1 |
| GPP | MODIS/061/MYD17A3HGF |
| AOD | MODIS/061/MOD08_M3 |
| Lai | MODIS/061/MCD15A3H |
| Fire | MODIS/061/MOD14A1 |

Date is the data name, and Source is the Source of the data obtained. GEE platform data source link: https://earthengine.google.com/

MOD16A2.061: It is an evapotranspiration and heat flux product with a temporal resolution of 8 days and a spatial resolution of 500 meters. In this study, the "ET" band of the "MODIS/006/MOD16A2" dataset in GEE is used to conduct statistics on ET data.

MODIS/061/MOD11A2: It is a LST product with a temporal resolution of 8 days and a spatial resolution of 500 meters. This study uses the "LST_Day_1km" band of the "MODIS/061/MOD11A2" dataset in GEE to calculate the LST data.

MOD17A2H: It is a gross primary productivity product with a temporal resolution of 8 days and a spatial resolution of 500 meters. In this study, the "Gpp" band of the "MODIS/006/MOD17A2H" dataset in GEE is used for the statistics of gross primary productivity.

MODIS/061/MOD08_M3: This data product contains atmospheric parameters related to atmospheric aerosol particle properties, total ozone load, atmospheric water vapor, cloud optical and physical properties, and atmospheric stability indices. This dataset plays an important role in studying atmospheric environmental changes.

MODIS/061/MCD15A3H: This data product includes LAI with a spatial resolution of 500 meters. The LAI variable is defined as the equivalent number of leaf layers per unit ground area. In this study, the LAI data from the MODIS/061/MCD15A3H dataset on the GEE platform is selected to study the influencing factors.

MOD14 is an important product of the MODIS (Moderate Resolution Imaging Spectroradiometer) fire detection and thermal anomaly dataset. It has a temporal resolution of daily and a spatial resolution of 1 km.

MODIS/061/MOD10A1 data product provides global snow cover information and belongs to the snow cover dataset with a spatial resolution of 500m. Snow cover is usually expressed as a percentage, indicating the proportion of snow in each pixel. Snow cover data has important applications in meteorology, climate research and other fields.

The vector boundaries of urban built-up areas were extracted from the China urban built-up areas 2020 dataset. Using ArcMap 10.8, the geographic coordinates (latitude and longitude) of each city's built-up area centroid were calculated using spatial analysis tools.

## Research methods

**Data preprocessing.** Remote sensing data processing: The geemap package was used to call and process datasets on the GEE platform in Python 3.8.

Sentinel-5P data processing: Remote sensing data vary in spatial and temporal resolution. Through the GEE platform, Sentinel-5P data were uniformly resampled and exported at a resolution of 1 kilometer. The processing chain involves converting L2 data into L3 data gridded by latitude and longitude using tools such as harpconvert, filtering out low-quality pixels, masking negative values, generating annual and quarterly averages, and performing statistics aggregation within built-up area boundaries.

Data preparation: The Pandas library in Python 3.8 was used for data organization and filtering, and the sklearn package was applied to impute missing values in the original dataset. This ensures the overall quality of the data and facilitates the establishment of a database.

**Statistical analysis.** This study examined $TNO_2CC$ across 346 major Chinese cities from 2019 to 2023. Raster datasets were homogenized to compute annual and seasonal averages, with seasons defined as follows: spring (March–May), summer (June–August), autumn (September–November), and winter (December–February). For the calculation of interannual mean values, we take into account that satellite observations are affected by factors such as cloud cover and ice/snow, resulting in uneven data availability across months. Therefore, when calculating seasonal mean values and annual mean values, we first compute the seasonal averages, and then use the arithmetic mean of these seasonal averages to derive the annual mean values. This approach ensures that each season contributes equally to the final annual result. ANOVA was applied to assess interannual differences in $TNO_2CC$, accompanied by characteristic maps illustrating inter annual and seasonal variation patterns.

**Spatial analysis.** Local spatial autocorrelation analysis: The local spatial autocorrelation method effectively detected spatial heterogeneity in urban atmospheric pollution patterns by identifying the geographic locations of pollution clusters and their aggregation types, such as high-high (HH) and low-low (LL) clusters. The calculation formulas were as follows:

$$I = \frac{(x_i - \bar{x})}{S^2} \sum_{j=1, j \neq i}^{n} W_{ij} (x_i - \bar{x}) \tag{1}$$

$$S^2 = \frac{1}{N} \sum_{i=1}^{n} (x_i - \bar{x})^2 \tag{2}$$

where $I$ represents the local spatial autocorrelation index, $x_i$ denotes the first element of the attribute, $\bar{x}$ represents the mean value of the nitrogen dioxide, $W_{ij}$ is the spatial weighting matrix, $S^2$ represents the variance of the attribute, $N$ is the number of elements.

Based on the local Moran's I index, this study used LISA maps to identify regional clustering patterns, categorizing spatial associations into four types: high-high (HH) clusters, low-low (LL) clusters, high-low (HL) spatial outliers, and low-high (LH) spatial outliers [31]. This methods enables the detection of statistically significant localized clusters and spatial anomalies, offering critical insights into the spatial heterogeneity of atmospheric $NO_2$ pollution across China's urban agglomerations [32–33]. LISA cluster analysis was further refined [34]. Each cluster type was defined as follows:

HH clusters: Areas where both the target region and its neighboring zones exhibit elevated $TNO_2CC$.

HL outliers: Areas with high $TNO_2CC$ levels surrounded by regions with low concentrations.

LH outliers: Areas with low $TNO_2CC$ adjacent to high-pollution neighbors.

LL clusters: Areas where both the target region and its surrounding area consistently demonstrate low $TNO_2CC$ levels.

Non-significant clusters: Regions lacking statistically significant spatial autocorrelation.

Subsequently, based on these four aggregation types, cities were classified and analyzed according to their temporal stability over five years period. This classification distinguished cities that consistently maintained a specific type of aggregation, those that changed once, and those that have undergone multiple transitions.

## The Random Forest (RF) model

The RF model is an ensemble learning algorithm [35], proposed by Breiman and Cutler in 2001, which uses decision trees as its base learners. During prediction, the RF algorithm employs the Bootstrap resampling method to draw samples from the original dataset. By iteratively constructing multiple decision trees through sampling with replacement, the model aggregates predictions from each tree and determines the final output via majority voting, thereby ensuring robust generalization performance through ensemble learning [36]. Compared to the traditional linear model, the RF model effectively captures complex interactions among various variables, offers fast training speeds, and does not require a predefined functional form [37]. Additionally, its simple structure and relatively few tuning parameters make it well-suited for multidimensional, multi-factor prediction tasks while delivering highly accurate prediction results [38].

Using the Random Forest (RF) model, follow these steps:

First, determine the `n_estimators` parameter. Then, further automate parameter tuning using grid search, setting the parameter tuning range.Adjust the `max_depth` parameter by establishing a tuning interval and using grid search for experimentation. When `max_depth` reaches the model's highest score, if this score is lower than when only `n_estimators` is set, the model should not use the `max_depth` parameter. Adjust the `min_samples_leaf` parameter with a defined tuning range and grid search. When `min_samples_leaf` achieves the model's highest score, if this score is lower than that of the model with only `n_estimators` set, the model should not use the `min_samples_leaf` parameter. Adjust the

`min_samples_split` parameter by setting a tuning interval and applying grid search. When `min_samples_split` reaches the model's highest score, if this score is lower than the model with only `n_estimators` configured, it indicates that adjusting `min_samples_split` can no longer optimize the model. Finally, adjust the `max_features` parameter using grid search. When `max_features` attains the model's highest score, the model reaches its optimal performance.

This study employed the PSO algorithm to optimize the hyperparameters of the RF model, facilitating an efficient exploration of the hyperparameter space to identify optimal configurations a. The RF model was trained using month average $TNO_2CC$ (2019–2023) and 15 explanatory variables.

**Shapley additive explanations.** SHAP is an additive explanation model that evaluates the impact of in-put variables on model predictions [39], SHAP quantifies the relative im-portance of input variables by assessing the average variation in model outputs due to changes in those variables [40], This is achieved through scatter plots and SHAP value distributions, which visualize variable contributions, model performance, and any biases in the estimates [41]. Applying SHAP values to interpret the optimized RF model provides deeper insights into the relative contributions of individual factors during training [42].

Assuming the *j*-th predictor variable of the *i*-th target variable is denoted as $x_{ij}$, the model's predicted value for the *i*-th target variable is $y_i$, and the average predicted value across all target variables is $y_{base}$, the SHAP value adheres to the following formula [43]:

$$y_i = y_{base} + f(x_{i1}) + f(x_{i2}) + f(x_{i3}) + ... + f(x_{iy}) \tag{3}$$

where $f(x_{ij})$ represents the SHAP value of the j-th predictor variable for the i-th target variable, indicating the marginal contribution of this predictor to the model's prediction of the target variable. In this study, the target variable was the $TNO_2CC$, with 15 explanatory variables as variable. Absolute SHAP values measure the magnitude of influence exerted by each predictor on the model's output, enabling variable importance ranking. A higher absolute SHAP value indicates a greater impact of the corresponding predictor on $TNO_2CC$ variability.

## Technical ideas

At present, many scholars use TROPOMI remote sensing inversion products to analyze and estimate the spatiotemporal concentration of air pollutants [44]. In this study, TROPOMI-drived $TNO_2CC$ data were used to explore its spatial and temporal distribution and natural influencing factors. The research consists of three main components, as shown in Fig 2:

First, using seasonal and annual time windows, statistical models such as analysis of variance (ANOVA) and Tamhane's T2 post-hoc tests were applied to analyze the temporal evolution patterns of $TNO_2CC$ across 346 major Chinese cities from 2019 to 2023. Second, local Moran's I and LISA cluster analysis were employed to identify high- and low-value spatial agglomerations and autocorrelation patterns.

$TNO_2CC$ (2019–2023) were used as the dependent variable, while 15 natural driving factors, including meteorological, vegetation, and anthropogenic indices, were used as variable. The model was optimized via Particle Swarm Optimization (PSO), resulting in the following hyperparameters: n_estimators = 4953, max_depth = 18, min_samples_split = 4, min_ samples_leaf = 1. By integrating the built-in feature importance ranking of the RF model with SHAP interpretation algorithms, this study identified key variables of air pollution. The top five natural influential factors were selected to generate partial dependence plots (PDPs), illustrating their nonlinear relationships with the dependent variable ($TNO_2CC$). Analyzing these relationships provides a basis for formulating effective urban air pollution prevention and control strategies in China.

This study used $TNO_2CC$ from 346 cities as the target variable. Based on existing research by domestic and international scholars on natural factors influencing atmospheric pollutant levels, 15 variables were selected, including Pressure, Lai, ET, and others, to train the RF model. Importance was ranked using the SHAP importance metric, quantified by the mean absolute SHAP value for each factor. Key natural drivers were subsequently identified, and their marginal effects on $TNO_2CC$ were analyzed through SHAP value decomposition, revealing nonlinear relationships and threshold behaviors in pollution dynamics.

## (a) Data acquisition and pre-treatment

## (b) Random Forest training

**Fig 2. Technology roadmap.** The administrative division data in GeoJSON format is sourced from the National Geospatial Information Public Service Platform (Tianditu), with the website: https://cloudcenter.tianditu.gov.cn/administrativeDivision. The map approval number is GS (2024) 0650. The data coordinates are in GCS_WGS_1984.

## Results

### Temporal distribution of TNO$_2$CC

The interannual variability characteristics of TNO$_2$CC in major Chinese cities are illustrated (Fig 3). From 2019 to 2023, the average annual NO$_2$ concentration exhibited a wave-shaped trend, initially increasing, then decreasing, and rising again. Between 2019 and 2021, TNO$_2$CC gradually increased, with a smaller difference between 2020 and 2021

compared to 2019. The average annual concentrations were $1.29 \times 10^{-4}$ mol/m², $1.42 \times 10^{-4}$ mol/m², and $1.47 \times 10^{-4}$ mol/m² for 2019, 2020, and 2021, respectively. The highest concentration over these three years occurred in 2021. In 2022, $NO_2$ concentration dropped significantly to $1.31 \times 10^{-4}$ mol/m². In 2023, concentrations increased slightly to $1.33 \times 10^{-4}$ mol/m².

This study applied ANOVA and Tamhane's T2 post-hoc tests to examine seasonal aver-ages of $TNO_2CC$ across major Chinese cities from 2019 to 2023. This approach enabled a rigorous comparison of interannual and intraseasonal variability in $NO_2$ pollution patterns, revealing statistically significant differences ($p < 0.05$) across climatic zones and urbanization gradients. The results in Table 2 confirm that significant differences in mean $TNO_2CC$ among the five years within each season. Notably, spring and summer exhibited relatively lower interannual variability, whereas winter demonstrated the most pronounced variability, possibly due to intensified heating emissions and stagnant meteorological conditions during the colder months, autumn is the season with the smallest difference in five years. The average annual concentration in autumn is the second highest value.

The seasonal variability of $TNO_2CC$ in China's major cities is depicted (Fig 4). The $TNO_2CC$ were ranked in descending order: winter > autumn > spring > summer. During 2019–2023, $TNO_2CC$ demonstrated significant seasonal variations, with peak concentrations observed in winter, lowest levels in summer, and moderate values in autumn and spring. The marginal difference between autumn and spring contrasted sharply with the winter-summer disparity, highlighting the dominant influence of seasonal emission patterns such as meteorological stagnation and precipitation is relatively low, making

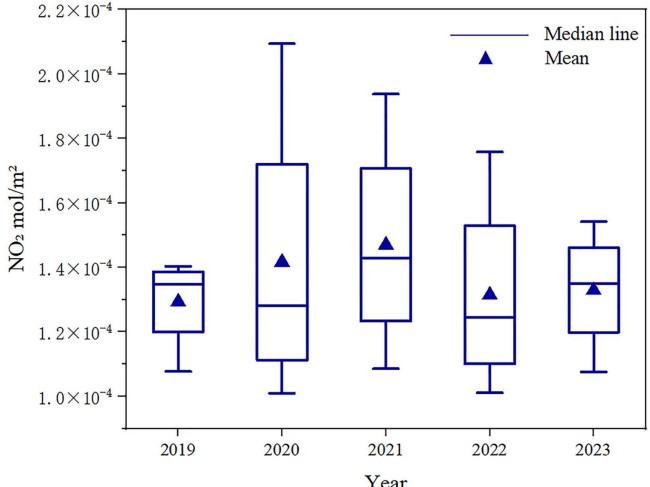

**Fig 3. Interannual variation characteristics of troposphere nitrogen dioxide vertical column density in major Chinese cities.**

**Table 2. Table of seasonal differences in five years.**

| Season | F | P |
|---|---|---|
| Spring | 30.66 | <0.05 |
| Summer | 16.76 | <0.05 |
| Autumn | 8.38 | <0.05 |
| Winter | 55.19 | <0.05 |

Season represents the name of each Season, P value (< 0.05) represents the difference between each Season, and F value represents the degree of seasonal difference between each year. The greater the F value, the greater the difference; On the contrary, the smaller the difference.

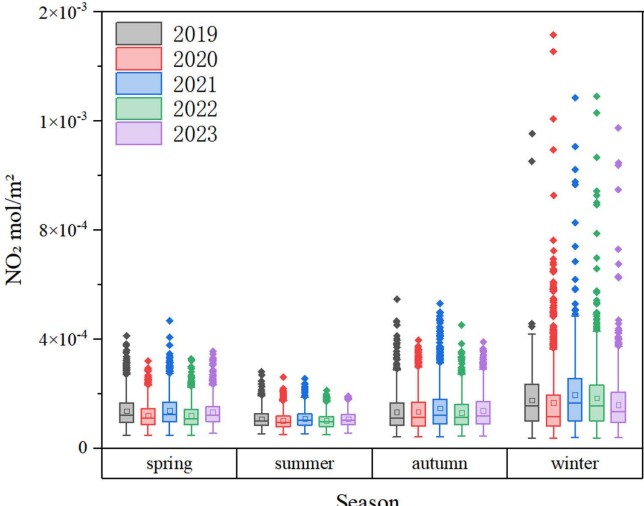

**Fig 4. Seasonal variation characteristics of vertical column density of nitrogen dioxide in the troposphere of major cities in China.** The points in the figure represent the annual average values of $TNO_2CC$ in each city; The square represents the mean value, and the horizontal line represents the median value.

it difficult for $NO_2$ pollutants to settle in winter. The figure shows more anomalies in winter, mainly due to seasonal effects, i.e., weaker photochemical sinks.

## Spatial distribution characteristics of $TNO_2CC$

A local spatial autocorrelation analysis model was applied to generate LISA cluster maps, illustrating the spatial heterogeneity of air pollution severity within the study area and its surrounding regions. Fig 5 shows the interannual variation in the number of cities belonging to each cluster types over the five-year period. The number of cities classified as HH and LL clusters remained relatively stable. In contrast, LH clusters exhibited moderate interannual fluctuations, with annual counts of 2, 1, 3, 5, and 2 cities, respectively.

This spatial pattern is influenced by Yulin's unique geographic location. Although local $TNO_2CC$ were relatively low, the city borders Shanxi Province to the east, a heavily industrialized region with elevated $TNO_2CC$ levels, and Yan'an City to the south. It is located in the northernmost part of Shaanxi and serves as a border area connecting five provinces and regions: Shanxi, Gansu, Ningxia, Inner Mongolia, and Shanxi. HL clusters were observed only in 2019, with Harbin City in Heilongjiang Province as the sole representative (Fig 6).

The administrative division data in GeoJSON format is sourced from the National Geospatial Information Public Service Platform (Tianditu), with the website: https://cloudcenter.tianditu.gov.cn/administrativeDivision. The map approval number is GS (2024) 0650. The data coordinates are in GCS_WGS_1984.

Significant spatial disparities in $TNO_2CC$ were evident across major Chinese cities (Fig 6), following a general decreasing gradient from east to west. High $TNO_2CC$ zones were primarily located east of the Hu Line (Hu Huanxian Line) and north of the Yangtze River Basin.

Elevated $TNO_2CC$ levels were observed in northern China. Similarly, high concentrations of $TNO_2CC$ were detected in the eastern coastal areas, particularly in the Yangtze River Delta and the Pearl River Delta regions. In Northeast China, significant $TNO_2CC$ accumulation was observed in metropolitan centers such as Harbin and Shenyang. Additionally, distinct regional hotspots of $TNO_2CC$ were identified in northern Xinjiang (including Urumqi) and within the Sichuan Basin. The highlands surrounding the Sichuan Basin hinder the horizontal and vertical dispersion of air pollutants, leading to the

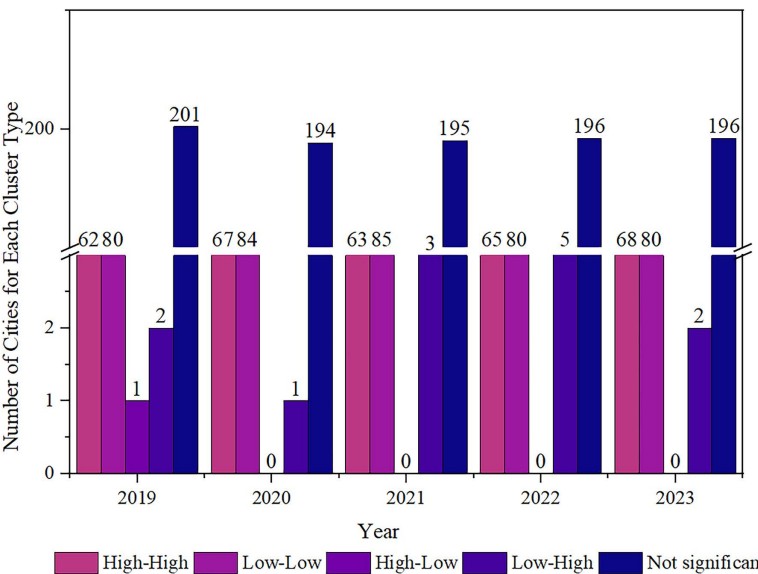

**Fig 5. Changes in the number of cities of each agglomeration type in five years.**

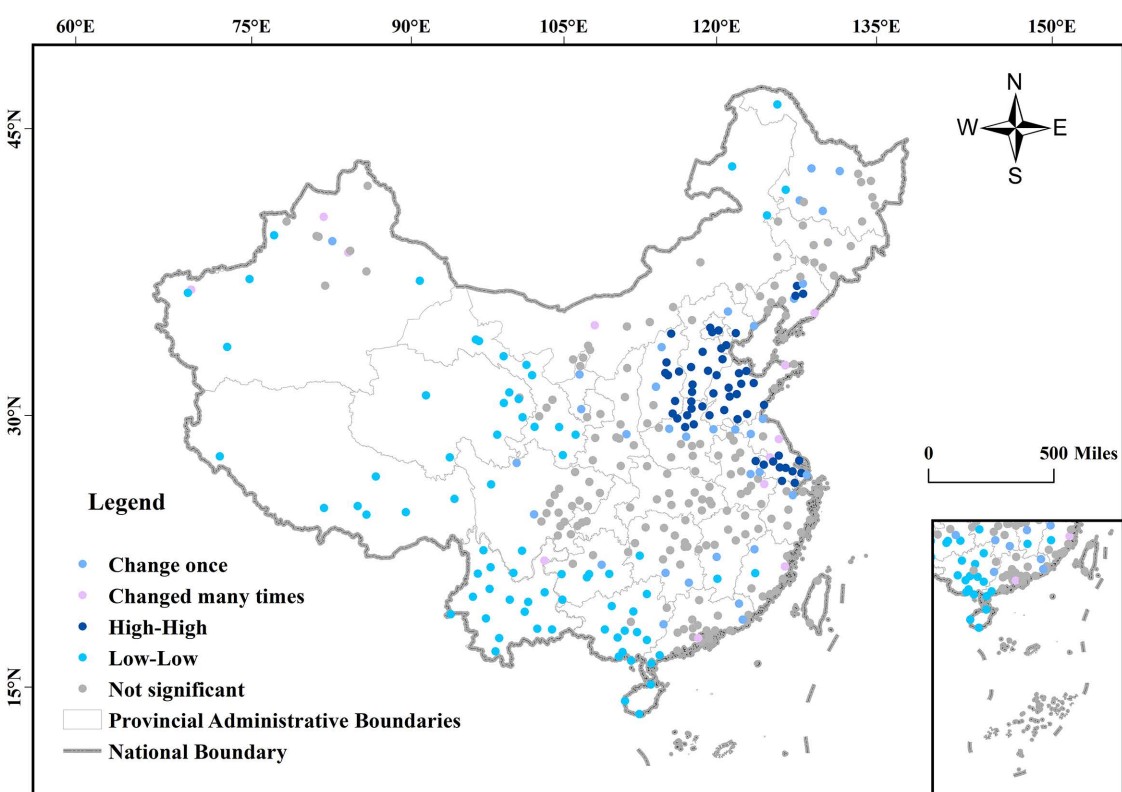

**Fig 6. Local spatial autocorrelation analysis of vertical column concentration of tropospheric nitrogen dioxide in major cities of China.**

accumulation of pollutants and thus an increase in TNO$_2$CC. This effect is particularly pronounced in provincial capitals like Chengdu.

Low TNO$_2$CC are mainly located south of the Yangtze River and west of the Hu Line, covering regions such as Northwest China (including southern Xinjiang, Ningxia, and Qinghai), Southwest China (including Tibet, Yunnan, and Guizhou), Southeast China (such as Guangdong and Fujian), and Northeast China (including Heilongjiang and eastern Inner Mongolia). The spatial distribution of TNO$_2$CC across China shows a distinct east-west gradient, with significantly higher concentrations in the east and lower levels in the west.

Over the past five years, 54 cities consistently remained classified as HH clustering areas. These were primarily concentrated in Beijing, Shijiazhuang, Jinan, Shanghai, Nanjing, and Hangzhou, as well as Shenyang and Benxi in Liaoning Province. Such regions, particularly the traditional industrial base in northeast China centered on Shenyang, have long maintained persistently high TNO$_2$CC. In contrast, 74 cities remained LL clusters, characterized by lower industrialization levels, moderate urbanization, and geographic isolation from high-emission zones, minimizing cross-regional pollution influence. Additionally, 34 cities transitioned once in cluster classification, including Qinhuangdao, Chengde, Heihe, and Luoyang. Among these, 20 cities, such as Nanping, Meizhou, and Daqing, exhibited relatively stable clustering patterns for 3–4 years, while 14 cities, including Ma'anshan, Jieyang, and Ya'an, exhibited non-significant clustering for 1–2 years but maintained a single cluster type in other years. A total of 12 cities, such as Fuzhou, Dandong, and Kizilsu Kirghiz Autonomous Prefecture, underwent multiple transitions in cluster classification. Notably, Xuancheng, Bayannur, Zhaotong, and four prefecture-level cities in Xinjiang, located near provincial capitals, and Zhongshan city near the Pearl River Delta, exhibited unstable clustering due to significant influence from adjacent high-emission zones. Coastal cities such as Fuzhou, Yancheng, Dandong, Yantai, and Yangzhou, which are situated along the Huaihe River's waterway, also exhibited heightened variability in cluster types. This variability was attributed to unstable meteorological conditions, including fluctuations in pressure, temperature, humidity, and WS.

From the perspective of climate zones, as shown in Fig 7, cities with high TNO$_2$CC are mainly concentrated in the II and III climate zones east of the Hu Huanyong Line, followed by the IV climate zone and the northern part of the V climate zone. Cities with low TNO$_2$CC are mainly distributed in the II and III climate zones west of the Hu Huanyong Line, as well as the I climate zone, the southern part of the V climate zone, the VI climate zone, and the VII climate zone.

## Natural factors influencing tropospheric TNO$_2$CC in Chinese cities

Using the processed month average TNO$_2$CC from 2019 to 2023, the following explanatory natural variables were selected: Pressure, Lai, Dew, AOD, WS, Month, ET, RH, Pre, Fire, LST, Temp, GPP, Year, and Snow. These variables formed the basis for constructing an optimized RF model to analyze the spatiotemporal drivers of TNO$_2$CC variability. Predictor importance was quantified using mean SHAP values, with the top five influential factors selected to generate PDPs. The RF model, combined with SHAP interpretation algorithms, facilitated an analysis of key drivers and used PDPs to isolate the marginal relationships between these natural factors and TNO$_2$CC while controlling for other variables. As shown in Fig 8, the results showed that the slopes of the modeling group and the validation group were 0.86 and 0.74, respectively, with R² values of 0.94 and 0.76. The data points of both the modeling group and the validation group were close to the 1:1 fitting line, as shown in Fig 8, indicating that the obtained RF model has a high degree of fitting.

Fig 9 illustrates the overall importance of each natural variable, with the y-axis representing ranked variable importance and the x-axis indicating mean SHAP values. The analysis demonstrated that Pressure exerted the strongest influence on TNO$_2$CC, followed by Lai, ET, Dew, and AOD. Based on this ranking, the top five factors, including Pressure, Lai, ET, Dew, and AOD, were identified as key natural drivers for in-depth analysis, while the remaining variables contributed minimally to the model's explanatory power (Fig 9). The parameter tuning results are as follows: S1 File.

The study revealed nonlinear relationships between these key drivers and TNO$_2$CC by applying PDPs to analyze the top five most influential factors. Pressure was identified as the most important natural variable. As Pressure increased,

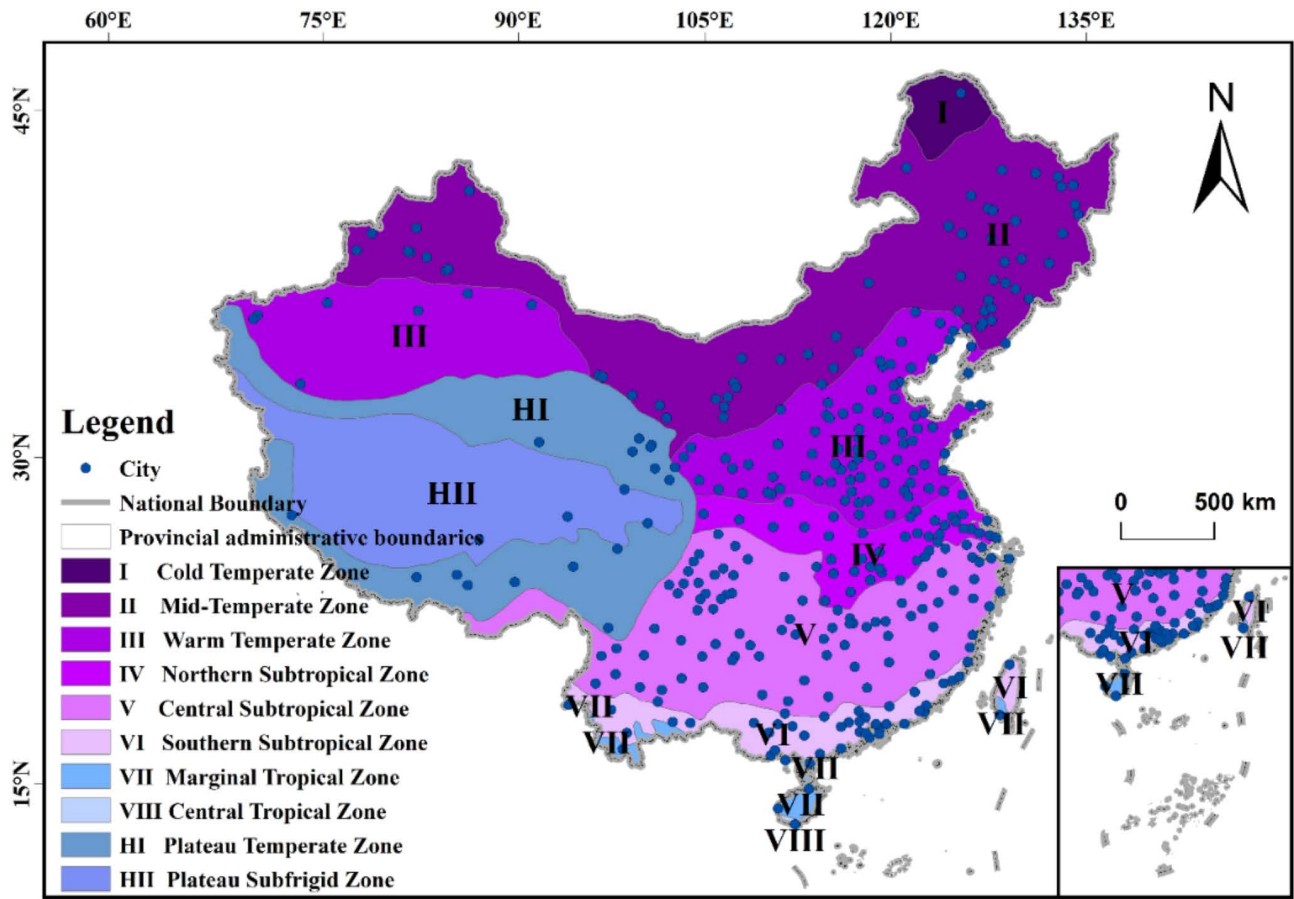

**Fig 7. Map of China's climate zoning.** The administrative division data in GeoJSON format is sourced from the National Geospatial Information Public Service Platform (Tianditu), with the website: https://cloudcenter.tianditu.gov.cn/administrativeDivision. The map approval number is GS (2024) 0650. The data coordinates are in GCS_WGS_1984.

$TNO_2CC$ exhibited an overall upward trend (Fig 10a). The relationship between Pressure and $TNO_2CC$ followed a wave-shaped pattern, rapidly escalating $TNO_2CC$ response magnitude when Pressure reached approximately 95,000 Pa. A pronounced positive correlation was observed in high-Pressure regions (> 95,000 Pa), indicating that elevated pressure consistently coincided with increased $TNO_2CC$ levels. The response of $TNO_2CC$ to AOD can be divided into two stages (Fig 10b). In the lower AOD range, $TNO_2CC$ sensitivity to AOD increased rapidly. Beyond an AOD value of approximately 330, the response gradually stabilized. $TNO_2CC$ participates in photochemical reactions in the atmosphere with other compounds, forming secondary aerosols. These secondary aerosols increase PM concentration, thereby elevating AOD values. The influence of Lai on $TNO_2CC$ exhibited a nonlinear trend, initially declining and then stabilizing (Fig 10c). Lower Lai values correspond to higher $TNO_2CC$, while higher Lai values correspond to low-er concentrations. ET exhibited a negative correlation with $TNO_2CC$ (Fig 10d). As ET increases, $TNO_2CC$ decreases, following an overall nonlinear trend of an initial sharp decline followed by gradual stabilization. Additionally, Dew also showed a negative correlation with $TNO_2CC$ overall, characterized by a brief increase followed by a continuous decrease, dividing the response into two phases (Fig 10e). Below approximately 263 K (low Dew range), Dew and $TNO_2CC$ exhibited a positive correlation. Above 263 K (high Dew range), the relationship shifts from positive to negative correlation, with $TNO_2CC$ gradually decreasing as Dew increases.

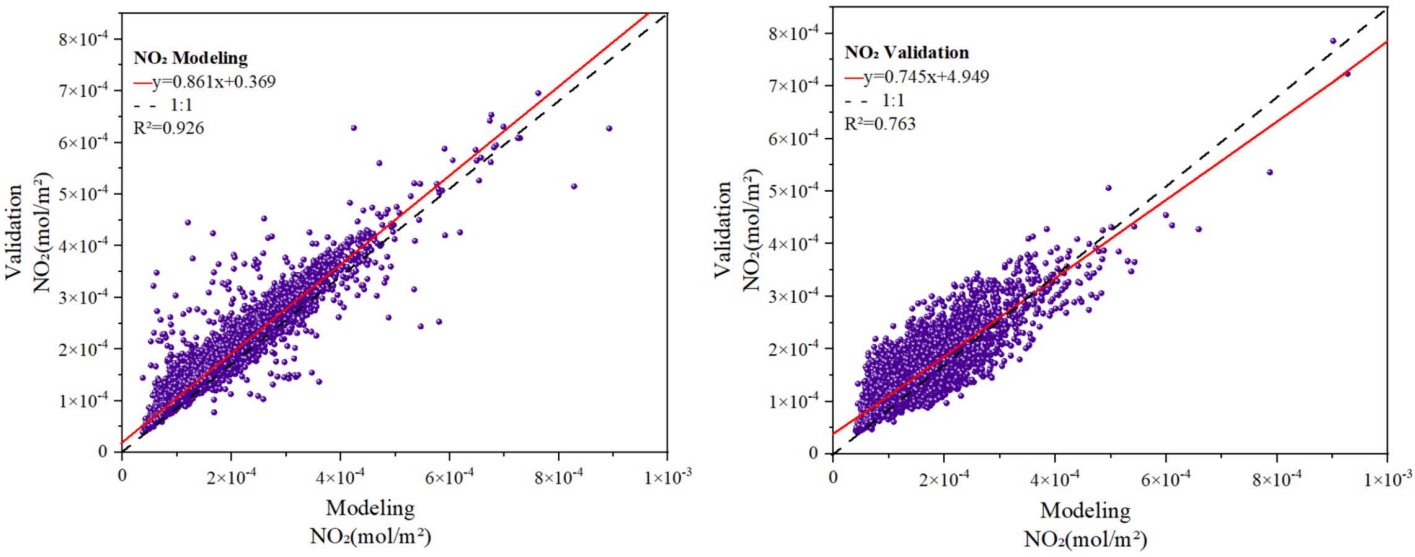

**Fig 8. Performance of the Random Forest (RF) model for nitrogen dioxide concentration.** (a): modeling, (b): validation.

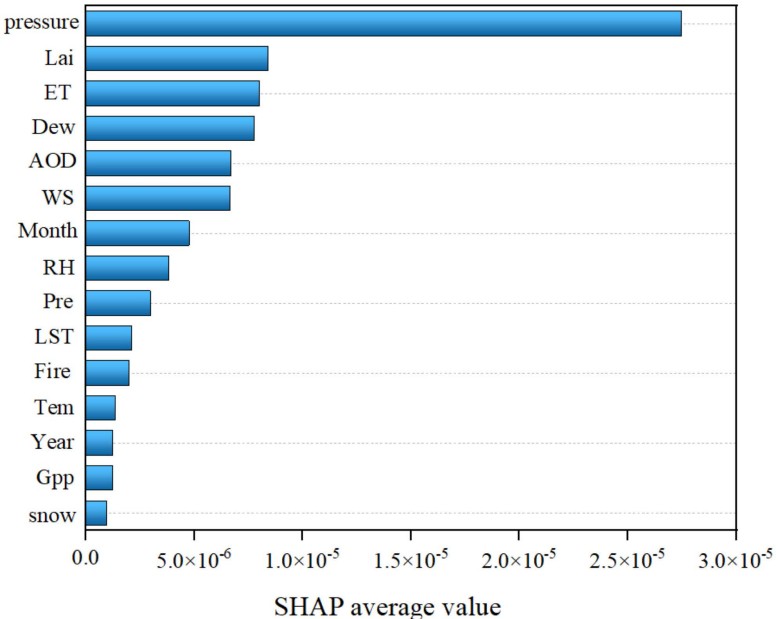

**Fig 9. Importance ranking of the affecting factors based on RF-SHAP model.**

## Discussion

A previous study applied the geographically weighted regression model to identify the the determinants of $PM_{2.5}$ concentration and explore variations in atmospheric pollutants [45]. Other research has employed principal component analysis, concluding that effective pollution control requires coordinated management of major pollutants, such as $PM_{10}$, $PM_{2.5}$, and $O_3$ [46]. The RF model offers several advantages, including low sensitivity to parameters [47], strong robustness

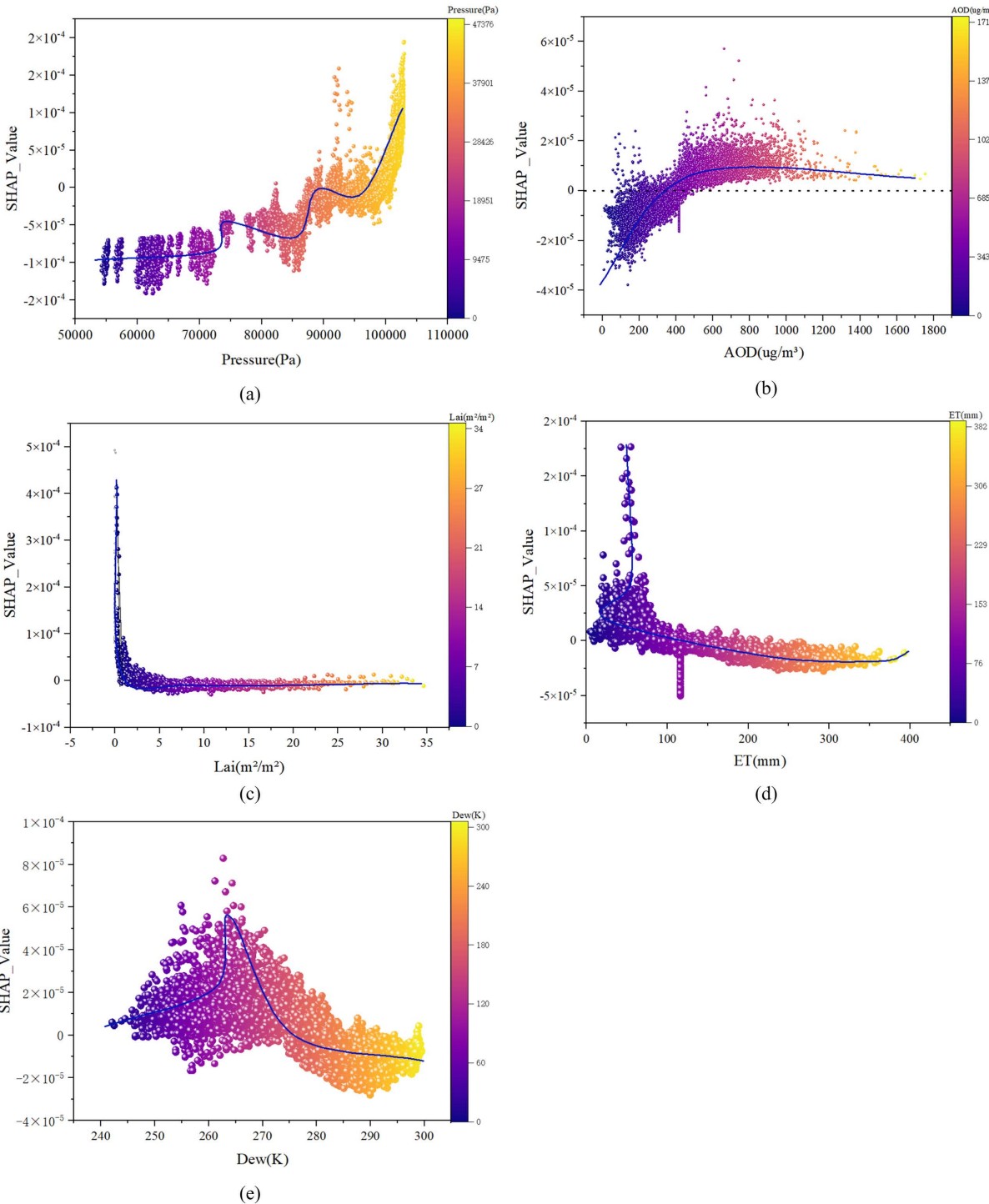

**Fig 10. Influence intensity of Pressure(a), AOD(b), Lai(c), ET(d), Dew(e) on nitrogen dioxide concentration.** (a): Pressure: atmospheric pressure; (b): AOD: aerosol optical depth; (c) Lai: Leaf Area Index; (d) ET: evapotranspiration; (e): Dew: dew point temperature. The blue line represents the trend of the impact of key influencing factors on TNO$_2$CC variation. SHAP values indicate the absolute effect size of features on the target variable, where the magnitude directly reflects each feature's marginal contribution to TNO$_2$CC variations. A SHAP value > 0 indicates a positive contribution, confirming that the factor's influence on TNO$_2$CC change increases as the SHAP value rises. Conversely, a SHAP value < 0 indicates a negative contribution, where a decreasing SHAP value corresponds to a stronger negative impact on TNO$_2$CC changes. Data on other influencing factors, except for the top five, can be found in S1 File.

against missing values, and improved computational efficiency through optimized variable selection [48]. The PSO algorithm exhibits excellent optimization performance [49]. This is particularly true in the field of environmental monitoring—for instance, in aspects such as pollutant concentration monitoring and water quality monitoring—where it demonstrates excellent performance [50]. This study employed an optimized RF model to identify key natural factors influencing TNO$_2$CC, providing theoretical foundations and data-driven insights for atmospheric pollution control.

This study analyzed correlation coefficients between natural variables to gain deeper insight in-to the factors and the mechanisms influencing TNO$_2$CC (Fig 11), the sample points comprised monthly average data from 346 cities spanning 2019–2023. Analysis of 15 natural variables determined the correlation analysis between variable pairs. The results indicated strong correlations (coefficients: > 0.65) between Lai and ET, GPP, Dew, Pre, LST, and Temp, whereas weak correlations (coefficients: > 0.65) were observed between AOD, Year, and other natural factors [51]. These findings suggest that the combined effects of vegetation and atmospheric processes mainly influence TNO$_2$CC changes.

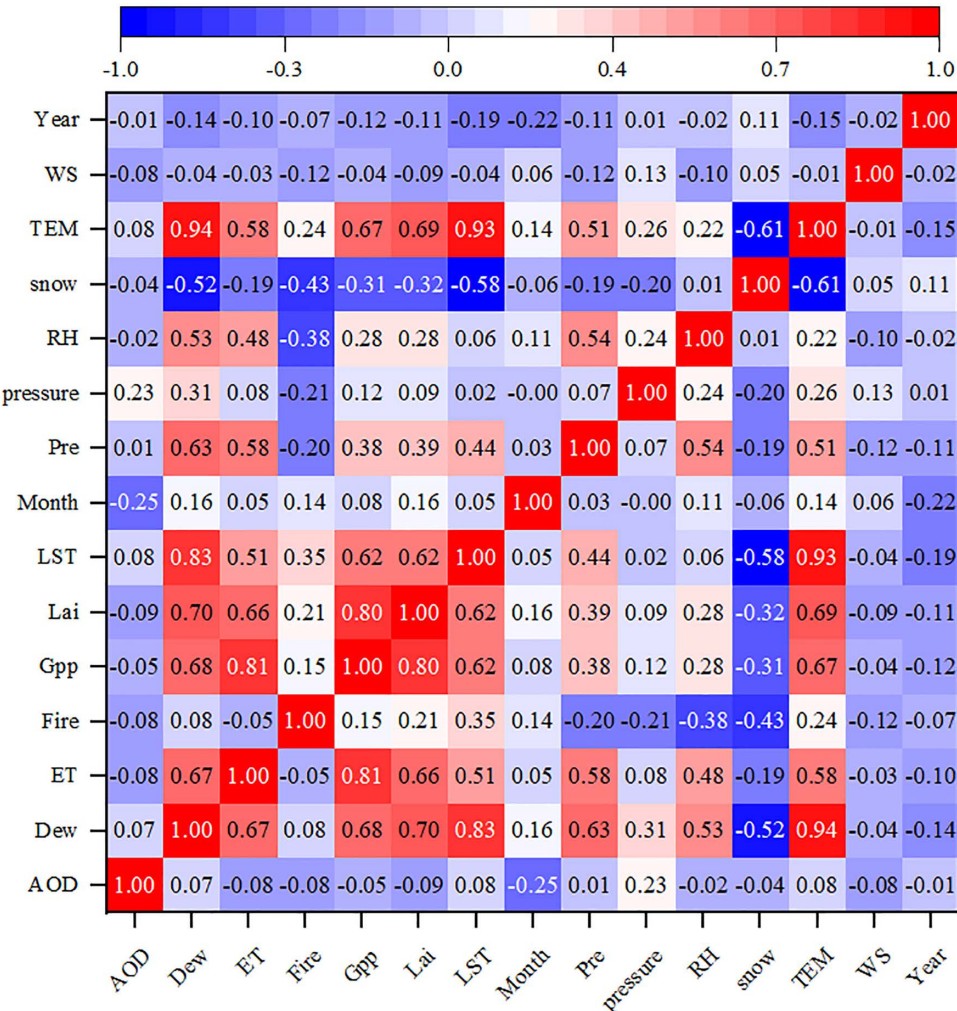

**Fig 11. Thermodynamic diagram of NO$_2$ influencing factors.** AOD (aerosol optical depth), Dew (dew point temperature), ET (evapotranspiration), Fire (fire activity), GPP (gross primary productivity), Lai (Leaf Area Index), LST (land surface temperature), Month(month), Pre (precipitation), Pressure (atmospheric pressure), RH (relative humidity), Snow (snow cover), TEMP (air temperature), WS (wind speed), Year (year).

The PSO algorithm was applied to optimize the RF model's hyperparameters and evaluate its robustness. Model performance was confirmed by calculating the $R^2$ using a linear regression model, indicating reasonable fitting accuracy (Fig 8). The built-in feature im-portance ranking function of the RF model was employed to prioritize 15 natural factors, including Pressure, Lai, Dew, ET, RH, AOD, Fire, GPP, Temp, Year, Month, Pre, LST, WS, and Snow, across the studied cities. Combined with the SHAP interpret ability algorithm (Fig 9). The analysis identified Pressure, AOD, Lai, ET, and Dew as the top five factors influencing $TNO_2CC$ variations, emphasizing the significant roles of pressure and vegetation in urban $NO_2$ pollutant dynamics. This finding contrasts with Wu et al who re-ported Temp and Pre as dominant natural drivers of $NO_2$ variability in the Shaanxi-Gansu-Ningxia region [52]. The discrepancy possibly arises from the region's unique topographic and climatic characteristics, highlighting the spatial heterogeneity of $NO_2$ pollution mechanisms.

$TNO_2CC$ exhibited strong seasonal variability, peaking in winter and declining in summer (Fig 4). This pattern is attributed to elevated pressure and lower temperatures in winter, which reduce the planetary boundary layer height and hinder pollutant dispersion [53]. Under such stable air conditions, $NO_2$ air pollutant accumulates [54]. Moreover, winter is characterized by static and stable weather conditions with low WS, leading to increased concentrations, whereas summer conditions promote better diffusion [55]. Although the annual average value in autumn is higher than that in summer and spring, the interannual difference is the smallest, as shown in Table 2. This phenomenon may occur because the meteorological conditions (such as WS, Temp and Pre) in autumn are more stable between years, so even if the $TNO_2CC$ is high, its interannual variation may be small. As shown in Fig 3, the interannual trend from 2019 to 2023 displays a wave-like pattern in annual average $TNO_2CC$. This pattern aligns with Zhang et al, who reported similar $TNO_2CC$ temporal variability in China, possibly influenced by prolonged high-Pressure systems during an extreme cold wave in December 2021 [56]. Notably, 2022 recorded the lowest annual $TNO_2CC$ within five years, followed by a rebound in 2023. Meteorological reports attribute the decline in 2022 to weaker cold air activity compared to previous years [57], collectively reducing emissions. In contrast, stronger cold air processes in 2023 may explain the renewed increase in $NO_2$ levels. Based on the analysis of climate zones, the impact of meteorological factors on the changes in $TNO_2CC$ has also been confirmed. The warm temperate zone features distinct four seasons, moderate precipitation, slightly cold winters and hot summers. In winter, static and stable weather conditions and temperature inversion phenomena often occur, making it difficult for near – surface pollutants (such as $NO_2$ and $PM_{2.5}$) to diffuse [58]. In summer, photochemical reactions and precipitation can remove part of the $NO_2$ in the atmosphere and reduce its concentration [59].

The model identified LAI, ET, and Dew as three of the top five natural influencing factors, all of which are vegetation-related variables. Correlation analysis underscores vegetation's role as a key regulator of $TNO_2CC$. LAI, defined as the total one-sided green leaf area per unit of ground surface [60], enhances photosynthetic efficiency, thereby reducing atmospheric $TNO_2CC$ through direct uptake and by facilitating wet deposition during Pre events [61]. Seasonally, higher LAI values in summer correspond to lower $TNO_2CC$ levels due to increased vegetation absorption and ET-driven rainfall. Spatially, southern China's dense vegetation exhibits stronger $NO_2$ removal capacity compared to northern regions, highlighting the importance of urban greening and vegetation coverage in shaping $TNO_2CC$ spatiotemporal patterns [62]. In addition to meteorological and vegetation factors, fire activity also have a certain impact on changes in $TNO_2CC$. However, according to the ranking results, the effect is relatively limited. Therefore, this factor will only cause a slight change in $TNO_2CC$ under special circumstances (such as urban fires) [63].

However, it is worth noting that this study focuses on the temporal-spatial distribution of $TNO_2CC$ and the research on its natural influencing factors. Nevertheless, it must be acknowledged that natural factors play a regulatory role in the influencing process, while human factors are the fundamental source [64]. This is an aspect that needs improvement in our future research—we should fully consider the impacts of other social factors. Nowadays, emission inventory data has become well-developed. Examples include the U.S. National Emission Inventory (NEI), the UK National Atmospheric Emission Inventory (NAEI), and China's Multi-resolution Emission Inventory for China (MEIC). Because of this, more researchers are using these emission inventories to analyze where pollutants come from [65–68]. In future work, we can

use these inventories to study the spatial distribution of air pollution sources. In terms of data validation, China's ground-based air quality monitoring stations are already quite comprehensive, and the temporal resolution of the measured data is also very high. Some researchers have already used ground monitoring data to verify the usability of S5P data from other countries [69–72]. Next, we can conduct ground-based validation research on China's S5P data to ensure the reliability of the data. Our goal is to provide a more scientific theoretical basis for air pollution control.

## Conclusions

This study analyzed the interannual and seasonal spatiotemporal distribution characteristics of $TNO_2CC$ in 346 major Chinese cities from 2019 to 2023, using satellite remote sensing data, statistical methods, and local spatial autocorrelation analysis. The key findings are summarized as follows:

(1) $TNO_2CC$ exhibited significant temporal heterogeneity across the five years. Annual trends demonstrated an initial rise, followed by a decline and rebound. Seasonally, concentration remained consistently higher in winter and lower in summer.

(2) $TNO_2CC$ across 346 major Chinese cities displayed notable spatial clustering. High-concentration hotspots were mainly distributed in urban areas of the North China Plain, Yangtze River Delta, and northeastern China. In contrast, low-concentration zones were concentrated in northwestern, southwestern, southern coastal, and Inner Mongolia regions. Additionally, monsoon climates in eastern China contribute to increased summer Pre, which dilutes $TNO_2CC$ levels, while centralized coal heating in northern regions during winters exacerbates $TNO_2CC$ accumulation, resulting in pronounced seasonal contrasts.

An RF model incorporating 15 natural influencing factors was optimized through iterative hyperparameter tuning, achieving high predictive accuracy with training and validation datasets closely aligned along the 1:1 fit line. Feature importance ranking using the SHAP algorithm identified Pressure, AOD, Lai, ET, and Dew as the top five drivers of $TNO_2CC$ variability. SHAP analysis revealed the following patterns: Pressure exhibited a positive correlation with $TNO_2CC$, following a wave-shaped upward trend. AOD transitioned from negative to positive effects on $TNO_2CC$ at thresholds of approximately 330. Lai and ET displayed negative correlations, with $TNO_2CC$ decreasing initially and stabilizing as these factors increased. Dew exerted a greater influence on $TNO_2CC$ near 263 K. Three different temperature measurements, Dew, LST, Temp. They themselves have a strong correlation. This shows that $TNO_2CC$ is strongly influenced by temperature.

## Supporting information

**S1 File. Results of random forest parameter tuning.** This text presents the optimal parameters obtained after tuning the random forest using the particle swarm optimization algorithm.
(PDF)

**S2 File. RF model parameter tuning code.** This document contains the code for the parameter tuning process of the Particle Swarm Optimization (PSO) algorithm.
(DOCX)

**S3 File. SHAP explanation algorithm code.** This document contains the code for the SHAP explanation algorithm, which is used to analyze the response degree of various influencing factors to $NO_2$ concentration.
(DOCX)

**S4 File. Raw data.**
(ZIP)

## Author contributions

**Conceptualization:** Feifei Cheng, Qing Sun, Heming Yang.

**Data curation:** Feifei Cheng, Qing Sun.

**Formal analysis:** Feifei Cheng, Qing Sun.

**Funding acquisition:** Heming Yang.

**Methodology:** Feifei Cheng, Qing Sun, Jiaqi Zhang, Jiahe Liu.

**Supervision:** Heming Yang.

**Visualization:** Feifei Cheng, Qing Sun, Jiaqi Zhang, Jiahe Liu, Wenwen Lü, Fapeng Shen.

**Writing – original draft:** Feifei Cheng, Qing Sun.

**Writing – review & editing:** Feifei Cheng, Qing Sun.

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
