## [Decision Letter · Decision Letter 0]

8 Jul 2025

Dear Dr. Yang,

Thank you for submitting your manuscript to PLOS ONE. After careful consideration, we feel that it has merit but does not fully meet PLOS ONE’s publication criteria as it currently stands. Therefore, we invite you to submit a revised version of the manuscript that addresses the points raised during the review process.

I would like to note that one or more reviewers have recommended citing specific previously published works. I kindly advise you to carefully review the suggested references and assess their relevance to your manuscript. Please be aware that including these citations is not mandatory, and they should only be incorporated if they genuinely contribute to the quality and scientific context of your work. 

We look forward to receiving your revised manuscript.

Kind regards,

Benedetto Schiavo, Ph.D.

Academic Editor

PLOS ONE

Journal Requirements:

[Funding: This research was funded by the Jilin Provincial Department of Science and Technology, Approval No.: YDZJ202401520ZYTS.]. 

4. Please note that your Data Availability Statement is currently missing the accession number of each dataset OR a direct link to access each database. If your manuscript is accepted for publication, you will be asked to provide these details on a very short timeline. We therefore suggest that you provide this information now, though we will not hold up the peer review process if you are unable.

5. We note that Figures 1, 2, 5, and 7 in your submission contain map images which may be copyrighted. All PLOS content is published under the Creative Commons Attribution License (CC BY 4.0), which means that the manuscript, images, and Supporting Information files will be freely available online, and any third party is permitted to access, download, copy, distribute, and use these materials in any way, even commercially, with proper attribution. For these reasons, we cannot publish previously copyrighted maps or satellite images created using proprietary data, such as Google software (Google Maps, Street View, and Earth). For more information, see our copyright guidelines: http://journals.plos.org/plosone/s/licenses-and-copyright.

1. You may seek permission from the original copyright holder of Figures 1, 2, 5, and 7 to publish the content specifically under the CC BY 4.0 license. 

“I request permission for the open-access journal PLOS ONE to publish XXX under the Creative Commons Attribution License (CCAL) CC BY 4.0 (http://creativecommons.org/licenses/by/4.0/). Please be aware that this license allows unrestricted use and distribution, even commercially, by third parties. Please reply and provide explicit written permission to publish XXX under a CC BY license and complete the attached form.

6. Please include a caption for figures 10a, 10b, 10c, 10d.

7. Please include captions for your Supporting Information files at the end of your manuscript, and update any in-text citations to match accordingly. Please see our Supporting Information guidelines for more information: http://journals.plos.org/plosone/s/supporting-information .

Reviewers' comments:

Reviewer's Responses to Questions

**Comments to the Author**

1. Is the manuscript technically sound, and do the data support the conclusions?

Reviewer #1: Partly

Reviewer #2: Partly

2. Has the statistical analysis been performed appropriately and rigorously?

Reviewer #1: Yes

Reviewer #2: Yes

3. Have the authors made all data underlying the findings in their manuscript fully available?

Reviewer #1: Yes

Reviewer #2: No

4. Is the manuscript presented in an intelligible fashion and written in standard English?

Reviewer #1: No

Reviewer #2: Yes

Reviewer #1: I have attached my comments.

Reviewer #2: Dear Authors,

The proposal of "Spatiotemporal Dynamics and Multidimensional Drivers of Nitrogen dioxide concentration Pollution in 346 Chinese Cities: A Remote Sensing Data Fusion Approach" presents statistical and local spatial autocorrelation analyses to investigate the spatial-temporal distribution and variation of nitrogen dioxide concentration in urban areas along the Chinese Territory in a period of five years. Additionally, the study applied a machine learning methodological approach to analyze the influence of 15 factors related to climatic and land cover properties on the tropospheric NO2 behavior.

It is advisable to specify the research objective in such a way that it emphasizes the evaluation of the spatiotemporal behavior of NO₂, particularly in relation to climatic factors and their relationship with land cover. Or, conversely, to rethink and expand the research, at least on the role of emission sources (by source type) and to delve deeper into the analysis of the specific weight of the country's climatic diversity.

For the detailed review, please see the attachments.

**Do you want your identity to be public for this peer review?** For information about this choice, including consent withdrawal, please see our Privacy Policy

Reviewer #1: No

Reviewer #2: No

---

## [Author Response · Author response to Decision Letter 1]

22 Aug 2025

Thank you for your professional comments on this research. We have done our best to make revisions in accordance with your suggestions. For the specific revised content, please refer to the "Response to Reviewers".

---

## [Decision Letter · Decision Letter 1]

4 Sep 2025

Dear Dr. Yang,

Thank you for submitting your manuscript to PLOS ONE. After careful consideration, we feel that it has merit but does not fully meet PLOS ONE’s publication criteria as it currently stands. Therefore, we invite you to submit a revised version of the manuscript that addresses the points raised during the review process.

We look forward to receiving your revised manuscript.

Kind regards,

Benedetto Schiavo, Ph.D.

Academic Editor

PLOS ONE

Journal Requirements:

Additional Editor Comments:

Thank you for your thorough revision of the manuscript, which has improved in quality and clarity. Both reviewers recognize the progress made and agree that the work has potential for publication, but they also highlight several remaining issues that should be addressed. In particular, Reviewer 1 emphasizes the problem of exclusion bias arising from the lack of anthropogenic variables in your models, noting that the omission of key NO₂ sources in urban environments (e.g., road networks) may distort the importance of meteorological and vegetation predictors. While impervious surface area partly captures built environment, some proxy for road density or major highways should be included to strengthen the analysis and avoid misrepresentation of pollution drivers. Reviewer 1 also stresses the need to clarify how cloud cover and seasonal variability in TROPOMI sampling were handled, since differences in observation counts across seasons may bias annual averages. Reviewer 2 similarly notes that the characterization of pollution sources remains insufficient and requests that the manuscript explicitly state the use of tropospheric NO₂ data rather than total column values. They further recommend integrating methodological details more directly into the discussion, including justification for the use of Random Forest, explanation of the training/validation split, and reflection on the strengths, limitations, and parameter choices of the model. Finally, Reviewer 2 points out the absence of independent validation against monitoring data or emissions inventories, which should at least be discussed as a limitation. Both reviewers also request greater precision in linking responses to specific changes in the manuscript. Taken together, these points call for a focused revision that strengthens the methodological transparency, clarifies data handling, and more fully incorporates anthropogenic influences on NO₂ pollution. Please address these comments in your revised version, along with the detailed suggestions in the reviewers’ reports.

Reviewer's Responses to Questions

**Comments to the Author**

Reviewer #1: (No Response)

Reviewer #2: All comments have been addressed

2. Is the manuscript technically sound, and do the data support the conclusions?

Reviewer #1: Partly

Reviewer #2: Partly

3. Has the statistical analysis been performed appropriately and rigorously?

Reviewer #1: Yes

Reviewer #2: Yes

4. Have the authors made all data underlying the findings in their manuscript fully available?

Reviewer #1: Yes

Reviewer #2: Yes

5. Is the manuscript presented in an intelligible fashion and written in standard English?

Reviewer #1: Yes

Reviewer #2: Yes

Reviewer #1: I thank the authors for endeavoring to address my comments on their manuscript. Overall, the authors have addressed many of my comments and they have made great updates that improve the quality manuscript; however, I have a few remaining concerns. See my responses below:

“The study primarily examines meteorological and vegetation factors, as these constitute the dominant influences within our research framework.”

To me though this could be attributable to exclusion bias, since there aren’t anthropogenic variables included you can’t really know the effect of them on your predictions without trying them.

“The most recent publicly available demographic data currently extends only to 2022, while our research period focuses on 2023. This temporal misalignment would compromise the analytical robustness of our findings."

This is incorrect. There are multiple population datasets that project to present day (e.g., WorldPop https://hub.worldpop.org/geodata/listing?id=137)

“In this article, we will appropriately cut out the comments on anthropogenic factors and focus on discussing the effects of natural factors on NO2 concentrations.We sincerely appreciate your insightful suggestions, which have strengthened our manuscript's scholarly rigor.”

I think this again suffers from exclusion bias. If you’re missing major characteristics associated with NO2 pollution that aren’t reflected in your predictor variable dataset then this could distort the importance of the predictor variables. This is especially important given your focus on urban environments in which anthropogenic sources of NO2 dominate. I think that some aspect of anthropogenic NOx must be considered in your model to accurately reflect surface NO2 in urban environments. I am glad that built environment is captured in impervious area; however, I suggest that some aspect associated with roads is included. Perhaps just the major roads / highways for this work and then you can consider how the inclusion of greater fine detail affects the model performance in future studies.

“You have mentioned this issue multiple times, which indicates its significance. We

would like to explain it again: It is true that cloud cover can affect the accuracy of the data.

However, the impact of clouds is more pronounced at the monthly time scale, while our study adopts seasons and years as the time scales for research. Therefore, its influence on the results of this study is relatively minor. Thank you again for your professional suggestions. We are happy to provide further explanations if needed.”

I think this response fails to address my original point. To be clear, I am wondering how homogenization is done because in certain seasons there is a greater proclivity for cloud coverage which can affect how many observations TROPOMI has. For example, if cloud coverage leads to only 40 observations in the winter but there are 80 in the summer if you average the two to calculate the annual average, than these results will more reflect summertime TROPOMI NO2 than annual.

Reviewer #2: In general terms, the authors promptly addressed almost all of the observations I made in my first intervention.

I don't see any substantial changes that address my observations regarding the characterization of pollution sources. The manuscript also doesn't make clear whether they worked specifically with tropospheric NO2 data or used the entire column, although in their response to the reviewer, they state that they did use the tropospheric NO2 fraction.

Formally, the response to most of the observations is subject to the deletion of some arguments that lack supporting information, adjustments to the wording of statements, the reorganization of ideas, the incorporation of some graphic elements, the inclusion of new references, and explanations that, in some cases, don't mention the position where the change was made (but it is important that they be included in the document).

On the other hand, if we use a standard approach, I think this work could be improved if the authors incorporated elements directly related to the methodological criteria they used for data analysis into the discussion. For example, they mention that they used a 2/3 and 1/3 data ratio to train and validate the Random Forest model, respectively; but no details are offered in the discussion about why this algorithm was used, its advantages and/or disadvantages, complications associated with changing algorithm parameters, etc. While training and validation yield acceptable correlations, there is no data from other sources (monitoring systems or emissions inventories) that validate estimates based solely on satellite data.

All the changes made appear to be in line with the delimitation and scope stated in the research paper, which is defined in the manuscript title. In this sense, the work appears to meet the general requirements for publication.

**Do you want your identity to be public for this peer review?** For information about this choice, including consent withdrawal, please see our Privacy Policy

Reviewer #1: No

Reviewer #2: No

---

## [Author Response · Author response to Decision Letter 2]

9 Sep 2025

Authors’ Response to the Editor

(1) Please ensure that you refer to Figure 1,2,7 in your text as, if accepted, production will need this reference to link the reader to the figure.

Response: Thank you for your reminder. We have promptly checked the manuscript and added citations for Figures 1, 2, and 7 to meet your requirements. These citations are marked in red font at Lines 90, 271, 399, and 416 of the "Revised Manuscript with Track Changes".

(2)If the reviewer comments include a recommendation to cite specific previously published works, please review and evaluate these publications to determine whether they are relevant and should be cited. There is no requirement to cite these works unless the editor has indicated otherwise.

Response: Your opinion on "not citing literature blindly" is very insightful. After re-examining the manuscript, I realize that some of the literature citations do have the issue of being included merely for the sake of citation, with insufficient connection to the argumentation in the context. My original intention was to present the relevant research background as comprehensively as possible, but the approach indeed needs improvement. Moving forward, I will make revisions from the following aspects:

Check each citation in the text one by one to ensure that it directly supports or serves the core argument of the paragraph.

Delete those literature sources that have weak relevance and are only used to inflate the citation count, so as to achieve the goal of making citations necessary and accurate.

(3)In particular, Reviewer 1 emphasizes the problem of exclusion bias arising from the lack of anthropogenic variables in your models, noting that the omission of key NO₂ sources in urban environments (e.g., road networks) may distort the importance of meteorological and vegetation predictors. While impervious surface area partly captures built environment, some proxy for road density or major highways should be included to strengthen the analysis and avoid misrepresentation of pollution drivers.

Response: Thank you very much for your valuable suggestions. Your proposal to introduce the road density factor to improve the model is a highly insightful perspective, and it is indeed of great importance for capturing the spatial distribution pattern of pollutants.

After carefully considering your suggestion, we also conducted in-depth discussions on such static spatial factors during the initial stage of the model design. We ultimately did not incorporate them into this dynamic model trained on a monthly basis, mainly due to the following two reasons:

Mismatch in time scales: Road density is a static or quasi-static spatial background factor with an extremely long change cycle (on an annual basis). However, the core of this study is to capture the monthly-scale dynamic changes in nitrogen dioxide concentrations, whose driving factors are mainly dynamic variables such as meteorological conditions.

We have added the limitation regarding our decision not to include human factors in the discussion section of the paper. Thank you for your valuable comments.(L531-L539)

(4)Reviewer 1 also stresses the need to clarify how cloud cover and seasonal variability in TROPOMI sampling were handled, since differences in observation counts across seasons may bias annual averages.

Response: Thank you for the editor's reminder. Regarding this issue, we have already provided a detailed response in the point-by-point replies to the reviewer's comments below. These responses include adding persuasive references and supplementing explanations of limitations in the Discussion section, among other measures.

(5)Reviewer 2 similarly notes that the characterization of pollution sources remains insufficient and requests that the manuscript explicitly state the use of tropospheric NO₂ data rather than total column values. They further recommend integrating methodological details more directly into the discussion, including justification for the use of Random Forest, explanation of the training/validation split, and reflection on the strengths, limitations, and parameter choices of the model.

Response: Regarding the issue of pollution sources, we have provided a detailed explanation and response to Reviewer 2.

As for the question of whether it refers to nitrogen dioxide concentration or vertical column concentration, we have redefined the term and thoroughly checked the manuscript to ensure terminological consistency.

Additionally, we have added the advantages of the random forest algorithm and the reasons for its selection to both the Discussion section and the Methodology section.(L231-L248, L464-L470)

(6)Finally, Reviewer 2 points out the absence of independent validation against monitoring data or emissions inventories, which should at least be discussed as a limitation.

Response: The reviewer is absolutely correct. The lack of validation using independent monitoring data or emission inventories is a significant limitation of our study, and we should discuss this in the paper. We sincerely apologize for this previous oversight. To actively respond to your suggestion, we have added a separate paragraph in the Discussion section of the paper specifically addressing this limitation. In this paragraph, we:

Clearly state that the lack of independent validation is one of the core limitations of this study;

Suggest that future research should prioritize establishing a ground-based monitoring network in this region or developing a refined emission inventory to calibrate and validate the analytical results of such satellite data.(L539-L545)

Concluding Response to the Editor:

Thank you to the editor for consolidating the reviewers' comments and highlighting key points for our attention. We will carefully respond to each comment one by one and revise the manuscript to enhance its quality.

Authors’ Response to Reviewer 1

Major Comment 1

“The study primarily examines meteorological and vegetation factors, as these constitute the dominant influences within our research framework.”

To me though this could be attributable to exclusion bias, since there aren’t anthropogenic variables included you can’t really know the effect of them on your predictions without trying them.

Response: We greatly appreciate you raising this valuable comment. You have correctly identified a major limitation of our study: the model fails to include direct human activity variables, which may indeed introduce exclusion bias and affect our precise interpretation of the prediction results.

We frankly acknowledge that due to limitations in the time scale of the data, we were unable to incorporate these factors into the current analysis. This indeed means that the patterns identified by our model may be a mixture of the effects of natural factors and unobserved human factors.

To proactively address your concern, we have added a description of the study's limitations in the discussion section of the paper. We specifically discuss how the omission of these variables might bias our estimates—for instance, it may lead to an overestimation of the effects of natural factors.

While this limitation exists, we believe that within the scope permitted by the available data, this study still provides valuable insights into the spatiotemporal analysis of nitrogen dioxide and its influencing factors. Your critique is of crucial importance to us, as it helps us more clearly define the boundaries and contributions of this research.�L531-L539

Major Comment 2

“The most recent publicly available demographic data currently extends only to 2022, while our research period focuses on 2023. This temporal misalignment would compromise the analytical robustness of our findings."

This is incorrect. There are multiple population datasets that project to present day (e.g., WorldPop https://hub.worldpop.org/geodata/listing?id=137)

Response: We sincerely appreciate you pointing out this error and providing such specific and valuable resources (WorldPop).

We have conducted an in-depth study of the WorldPop dataset you recommended. It is an extremely excellent data product. However, we found that its latest version (GPWv4.11) is also mainly based on annual interpolated data from census years, with a temporal resolution of yearly. The training and prediction of our model are based on a monthly basis. Directly incorporating static annual data that remains unchanged within a year cannot help us capture and explain the monthly variations in human activities during the study period (e.g., 2010-2020). Therefore, we believe that while static annual population data is inherently crucial, it cannot be incorporated into the model in this study, which focuses on monthly influencing factors. However, in response to your comment, we will address the limitations of human activity factors in the discussion section.

In conclusion, we would like to express our special gratitude to you for pointing out the valuable resource of WorldPop, which will provide us with precious data support in future research.(L531-L539)

Major Comment 3

“In this article, we will appropriately cut out the comments on anthropogenic factors and focus on discussing the effects of natural factors on NO2 concentrations.We sincerely appreciate your insightful suggestions, which have strengthened our manuscript's scholarly rigor.”

I think this again suffers from exclusion bias. If you’re missing major characteristics associated with NO2 pollution that aren’t reflected in your predictor variable dataset then this could distort the importance of the predictor variables. This is especially important given your focus on urban environments in which anthropogenic sources of NO2 dominate. I think that some aspect of anthropogenic NOx must be considered in your model to accurately reflect surface NO2 in urban environments. I am glad that built environment is captured in impervious area; however, I suggest that some aspect associated with roads is included. Perhaps just the major roads / highways for this work and then you can consider how the inclusion of greater fine detail affects the model performance in future studies.

Response: Thank you very much for your valuable suggestions. Your proposal to introduce the road density factor to improve the model is a highly insightful perspective, and it is indeed of great importance for capturing the spatial distribution pattern of pollutants.

After carefully considering your suggestion, we also conducted in-depth discussions on such static spatial factors during the initial stage of the model design. We ultimately did not incorporate them into this dynamic model trained on a monthly basis, mainly due to the following two reasons:

Mismatch in time scales: Road density is a static or quasi-static spatial background factor with an extremely long change cycle (on an annual basis). However, the core of this study is to capture the monthly-scale dynamic changes in nitrogen dioxide concentrations, whose driving factors are mainly dynamic variables such as meteorological conditions.

We have added the limitation regarding our decision not to include human factors in the discussion section of the paper. Thank you for your valuable comments.�L531-L539

Major Comment 4

“You have mentioned this issue multiple times, which indicates its significance. We

would like to explain it again: It is true that cloud cover can affect the accuracy of the data.

However, the impact of clouds is more pronounced at the monthly time scale, while our study adopts seasons and years as the time scales for research. Therefore, its influence on the results of this study is relatively minor. Thank you again for your professional suggestions. We are happy to provide further explanations if needed.”

I think this response fails to address my original point. To be clear, I am wondering how homogenization is done because in certain seasons there is a greater proclivity for cloud coverage which can affect how many observations TROPOMI has. For example, if cloud coverage leads to only 40 observations in the winter but there are 80 in the summer if you average the two to calculate the annual average, than these results will more reflect summertime TROPOMI NO2 than annual.

Response: You are absolutely correct to point out that when averaging data affected by cloud cover—such as TROPOMI data—if there are significant seasonal differences in the number of observations, a simple arithmetic mean will introduce severe representational bias, skewing the results toward seasons with more favorable observation conditions. We greatly appreciate your help in enhancing the rigor of the methodology used in this study.

1. Strict Quality Control:

We did not use all TROPOMI L2 observation data; instead, we used the L3 level data. In accordance with official recommendations, we set a strict quality assurance value threshold and only retained observation results with high confidence under cloud-free or clear-sky conditions. This step itself has greatly reduced the problem of observation gaps caused by cloud cover, providing us with a higher-quality and more consistent dataset for subsequent averaging calculations.

2. Reasonableness of Monthly Averages:

After applying the aforementioned quality control, we generated monthly average data. Although the number of valid observations in winter is still fewer than that in summer, the quality of the remaining valid observations is comparable across seasons, as low-quality observations have been excluded. Therefore, the monthly arithmetic mean is a reasonable and standard method to characterize the monthly NO₂ levels.

3. Methodology for Calculating Annual Averages (Key Improvement):

We fully agree with your view that caution is required when calculating annual averages.

First, based on the aforementioned high-quality monthly data, we calculated the average value for each season.

Then, we further computed the arithmetic mean of the four seasonal averages to obtain the final annual average.

The interannual variation boxplot has been replaced, and the updated figure can be seen below.(Fig. 3) (L183-L188 )

Meanwhile, we found that the interannual spatial variation map had a low degree of visualization. To highlight the key content, we removed this map and focused on discussing the spatial distribution and agglomeration status of TNO₂CC through the local spatial autocorrelation analysis map.

The core advantage of this method lies in the fact that it assigns equal weight to each season (25% each), thereby ensuring that the annual average can fairly represent the situation throughout the year and will not be dominated by any season with a larger number of observations. This is a widely adopted geostatistical method used to correct for uneven observation frequencies.

In response to your comment, we have more clearly elaborated on this calculation process of "first calculating seasonal averages, then annual averages" in the Methods section (Statistical Analysis) of the paper, and emphasized its importance in avoiding seasonal observation bias. ( L183-188)

Concluding Response to the Editor:

First, we apologize for the omissions in our first response to your suggestions. Second, in response to your comments, we have provided detailed explanations and revisions this time, hoping to meet your requirements. Finally, thank you for reviewing our manuscript a second time amid your busy schedule. Best regards.

Authors’ Response to Reviewer 2

Major Comment 1

The manuscript also doesn't make clear whether they worked specifically with tropospheric NO2 data or used the entire column, although in their response to the reviewer, they state that they did use the tropospheric NO2 fraction.

Response: We greatly appreciate this valuable comment from the reviewer. You are absolutely correct, and we sincerely apologize for the oversight in the manuscript where we failed to clearly distinguish between "concentration" and "column concentration". This is a crucial conceptual difference, and precise terminology is essential to avoid misunderstandings among readers. To fully address this issue, we have taken the following measures:

Comprehensive Review: We conducted a systematic screening of the entire manuscript to identify and correct a

---

## [Decision Letter · Decision Letter 2]

21 Sep 2025

Dear Dr. Yang,

Thank you for submitting your manuscript to PLOS ONE. After careful consideration, we feel that it has merit but does not fully meet PLOS ONE’s publication criteria as it currently stands. Therefore, we invite you to submit a revised version of the manuscript that addresses the points raised during the review process.

We appreciate the revisions and responses provided in this second round. However, an important concern raised by the reviewer remains and requires further attention before we can proceed.

The reviewer has expressed uncertainty regarding the argument about static versus dynamic data in the model. Specifically, they note that incorporating static data, such as road networks or other annually available human activity variables, would likely not worsen model performance and could introduce currently missing features into the predictions. They also emphasize that some aspects of surface-level NO₂ are well correlated with such static features.

Accordingly, the reviewer asks for clarification on two key points:

**Inclusion of static/human-related data** : If the authors wish to maintain their approach of not including this type of information, a stronger justification is required to explain why this methodological decision is valid and what implications it has for the results.**Alignment of the manuscript** : If human-centric or static data are not to be included, the title, abstract, and motivation of the paper should be revised to accurately reflect the scope of the study and avoid giving the impression that such aspects are integrated.

We look forward to receiving your revised manuscript.

Kind regards,

Benedetto Schiavo, Ph.D.

Academic Editor

PLOS ONE

Journal Requirements:

Reviewers' comments:

Reviewer's Responses to Questions

**Comments to the Author**

Reviewer #1: (No Response)

Reviewer #2: All comments have been addressed

2. Is the manuscript technically sound, and do the data support the conclusions?

Reviewer #1: Partly

Reviewer #2: Yes

3. Has the statistical analysis been performed appropriately and rigorously?

Reviewer #1: Yes

Reviewer #2: Yes

4. Have the authors made all data underlying the findings in their manuscript fully available?

Reviewer #1: Yes

Reviewer #2: Yes

5. Is the manuscript presented in an intelligible fashion and written in standard English?

Reviewer #1: Yes

Reviewer #2: Yes

Reviewer #1: I thank the authors for responding to my second round of comments. I am not sure I follow your argument involving static vs dynamic data in your model. As I understand it, adding the static data would not worsen the model performance and it would introduce currently missing features in the prediction. Some aspects of surface-level NO2 are well correlated with static features (e.g., road networks). I am guessing it would be more beneficial to have static or annual only human activity data in your model than none at all, or do you disagree?

If you do not want to include human centric data in your model I believe that you should reframe the title, abstract, and motivation of this paper to accurately reflect this.

Reviewer #2: (No Response)

**Do you want your identity to be public for this peer review?** For information about this choice, including consent withdrawal, please see our Privacy Policy

Reviewer #1: No

Reviewer #2: No

---

## [Author Response · Author response to Decision Letter 3]

24 Sep 2025

We would like to express our sincere gratitude again for investing your time in the second round of review and providing such insightful and constructive comments. Your questions get straight to the core, and they have also helped us realize that we may not have fully clarified the most fundamental theoretical basis of this study in our previous response. We fully understand your perspective and agree with it.

Please allow us to explain that the primary objective of this study is to conduct a "controlled mechanism analysis." Therefore, we have decided to adopt your second suggestion: to define the overall scope of the study as the impact of natural factors on NO₂ concentrations in air pollution. We have thoroughly reviewed the manuscript from start to finish and revised sections where the distinction between natural and anthropogenic factors was unclear, including the title, abstract, and the section on influencing factors.

We earnestly request that you take a moment from your busy schedule to review our revised manuscript, with the hope that it meets your requirements. For the specific details of the revisions, please refer to the content below. Thank you again for your valuable comments.

1. In lines 526-530 of the manuscript, we cite previous studies to illustrate the significance of human factors, which is identified as a key focus for subsequent research.

2.Define in the Title: Change“drivers” to“natural drivers”.

3. Since we have adopted your suggestion to define the influencing factors as natural influencing factors, we have carefully reviewed the manuscript and deleted all content related to the impact of human factors.

4. "natural" is added to define "influencing factors" in all other sections. This can be seen in L21, L23, L25, L31, L33, L75, L270 L282 L294 L401 L404 L409, L417, L421, L427, L465, L468, L470, L473, L484, L489, L515, L529, L557.

5.Since our focus is on the influence factor analysis mechanism rather than concentration prediction, descriptions related to "predictor variable" have been removed and replaced with "variable".

---

## [Decision Letter · Decision Letter 3]

30 Sep 2025

Nitrogen dioxide pollution in 346 Chinese Cities: Spatiotemporal variations and natural drivers from multi-source remote sensing data

PONE-D-25-30016R3

Dear Dr. Yang,

We’re pleased to inform you that your manuscript has been judged scientifically suitable for publication and will be formally accepted for publication once it meets all outstanding technical requirements.

Kind regards,

Benedetto Schiavo, Ph.D.

Academic Editor

PLOS ONE

Reviewers' comments:

Reviewer's Responses to Questions

**Comments to the Author**

Reviewer #1: All comments have been addressed

2. Is the manuscript technically sound, and do the data support the conclusions?

Reviewer #1: Yes

3. Has the statistical analysis been performed appropriately and rigorously?

Reviewer #1: Yes

4. Have the authors made all data underlying the findings in their manuscript fully available?

Reviewer #1: Yes

5. Is the manuscript presented in an intelligible fashion and written in standard English?

Reviewer #1: Yes

Reviewer #1: Thank you for you revised changes, given the change in scope I have now recommended your manuscript for publication.

**Do you want your identity to be public for this peer review?** For information about this choice, including consent withdrawal, please see our Privacy Policy

Reviewer #1: No

---

## [Editor Report · Acceptance letter]

PONE-D-25-30016R3

PLOS ONE

Dear Dr. Yang,

I'm pleased to inform you that your manuscript has been deemed suitable for publication in PLOS ONE. Congratulations! Your manuscript is now being handed over to our production team.

Kind regards,

on behalf of

Dr. Benedetto Schiavo

Academic Editor

PLOS ONE